# The Entropy Universe

**DOI:** 10.3390/e23020222

**Published:** 2021-02-11

**Authors:** Maria Ribeiro, Teresa Henriques, Luísa Castro, André Souto, Luís Antunes, Cristina Costa-Santos, Andreia Teixeira

**Affiliations:** 1Institute for Systems and Computer Engineering, Technology and Science (INESC-TEC), 4200-465 Porto, Portugal; lfa@fc.up.pt; 2Computer Science Department, Faculty of Sciences, University of Porto, 4169-007 Porto, Portugal; 3Centre for Health Technology and Services Research (CINTESIS), Faculty of Medicine University of Porto, 4200-450 Porto, Portugal; teresasarhen@med.up.pt (T.H.); luisacastro@med.up.pt (L.C.); csantos@med.up.pt (C.C.-S.); andreiasofiat@med.up.pt (A.T.); 4Department of Community Medicine, Information and Health Decision Sciences-MEDCIDS, Faculty of Medicine, University of Porto, 4200-450 Porto, Portugal; 5LASIGE, Faculdade de Ciências da Universidade de Lisboa, 1749-016 Lisboa, Portugal; ansouto@fc.ul.pt; 6Departamento de Informática, Faculdade de Ciências da Universidade de Lisboa, 1749-016 Lisboa, Portugal; 7Instituto de Telecomunicações, 1049-001 Lisboa, Portugal; 8Instituto Politécnico de Viana do Castelo, 4900-347 Viana do Castelo, Portugal

**Keywords:** entropy measures, information theory, time-series, application areas

## Abstract

About 160 years ago, the concept of entropy was introduced in thermodynamics by Rudolf Clausius. Since then, it has been continually extended, interpreted, and applied by researchers in many scientific fields, such as general physics, information theory, chaos theory, data mining, and mathematical linguistics. This paper presents *The Entropy Universe*, which aims to review the many variants of entropies applied to time-series. The purpose is to answer research questions such as: How did each entropy emerge? What is the mathematical definition of each variant of entropy? How are entropies related to each other? What are the most applied scientific fields for each entropy? We describe in-depth the relationship between the most applied entropies in time-series for different scientific fields, establishing bases for researchers to properly choose the variant of entropy most suitable for their data. The number of citations over the past sixteen years of each paper proposing a new entropy was also accessed. The Shannon/differential, the Tsallis, the sample, the permutation, and the approximate entropies were the most cited ones. Based on the ten research areas with the most significant number of records obtained in the Web of Science and Scopus, the areas in which the entropies are more applied are computer science, physics, mathematics, and engineering. The universe of entropies is growing each day, either due to the introducing new variants either due to novel applications. Knowing each entropy’s strengths and of limitations is essential to ensure the proper improvement of this research field.

## 1. Introduction

Despite its long history, to many, the term entropy still appears not to be easily understood. Initially, the concept was applied to thermodynamics, but it is becoming more popular in other fields. The concept of entropy has a complex history. It has been the subject of diverse reconstructions and interpretations making it very confusing and difficult to understand, implement, and interpret.

Up to the present, many different types of entropy methods have emerged, with a large number of different purposes and possible application areas. Various descriptions and meanings of entropy are provided in the scientific community, bewildering researchers, students, and professors [1,2,3]. The miscellany in the research papers by the widespread use of entropy in many disciplines leads to many contradictions and misconceptions involving entropy, summarized in Von Neumann’s sentence, *“Whoever uses the term ‘entropy’ in a discussion always wins since no one knows what entropy really is, so in a debate, one always has the advantage”* [4,5].

Researchers have already studied entropy measurement problems, but there are still several questions to answer. In 1983, Batten [6] discussed entropy theoretical ideas that have led to the suggested nexus between the physicists’ entropy concept and measures of uncertainty or information. Amigó et al. [7] presented a review of only generalized entropies, which from a mathematical point of view, are non-negative functions defined on probability distributions that satisfy the first three Shannon–Khinchin axioms [8]. In 2019, Namdari and Zhaojun [9] reviewed the entropy concept for uncertainty quantification of stochastic processes of lithium-ion battery capacity data. However, those works do not present an in-depth analysis of how entropies are related to each other.

Several researchers, such as the ones of reference [10,11,12,13,14,15,16,17,18,19], consider that entropy is an essential tool for time-series analysis and apply this measure in several research areas. Nevertheless, the choice of a specific entropy is made in an isolated and unclear way. There is no complete study on the areas of entropies application in the literature to the best of our knowledge. We believe that a study of the importance and application of each entropy in time-series will help researchers understand and choose the most appropriate measure for their problem.

Hence, considering entropies applied to time-series, the focus of our work is to demystify and clarify the concept of entropy, describing:How the different concepts of entropy arose.The mathematical definitions of each entropy.How the entropies are related to each other.Which are the areas of application of each entropy and their impact in the scientific community.

## 2. Building the Universe of Entropies

In this section, we describe in detail how we built our universe of entropies. We describe the entropies, mathematical definitions, the respective origin, and the relationship between each other. We also address some issues with the concept of entropy and, its extension to the study of continuous random variables.

Figure 1 is the timeline (in logarithmic scale) of the universe of entropies covered in this paper. Although we have referred to some entropies in more than one section, in the Figure 1, the colors refer to the section where each entropy has been defined.

Boltzmann, Gibbs, Hartley, quantum, Shannon and Boltzmann-Gibbs-Shannon entropies, the first concepts of entropy are described in Section 2.1. Section 2.2 is dedicated to entropies derived from Shannon entropy such as differential entropy (Section 2.2.1), spectral entropy (Section 2.2.2), tone-entropy (Section 2.2.3), wavelet entropy (Section 2.2.4), empirical mode decomposition energy entropy (Section 2.2.5), and Δ-entropy (Section 2.2.6).

Kolmogorov, topological, and geometric entropies are described in Section 2.3. Section 2.4 describes the particular cases of Rényi entropy (Section 2.4.1), ϵ−smooth Rényi entropy (Section 2.4.2), and Rényi entropy for continuous random variable and the different definition of quadratic entropy (Section 2.4.3). Havrda–Charvát structural α-entropy and Tsallis entropy are detailed in Section 2.5 while permutation entropy and related entropies are described in Section 2.6.

Section 2.7 is dedicated to rank-based and bubble entropies while topological information content, graph entropy and horizontal visibility graph entropy are detailed in Section 2.8. In Section 2.9,are described the approximate, the sample, and related entropies such as the quadratic sample, coefficient of the sample, and intrinsic mode entropies (Section 2.9.1), dispersion entropy and fluctuation-based dispersion entropy (Section 2.9.2), fuzzy entropy (Section 2.9.3), modified sample entropy (Section 2.9.4), fuzzy measure entropy (Section 2.9.5), and kernel entropies (Section 2.9.6).

### 2.1. Early Times of the Entropy Concept

In 1864, Rudolf Clausius introduced the term entropy in thermodynamics from the Greek word *Entropein* for transformation and change [20]. The concept of entropy arises, providing a statement of the second law of thermodynamics. Later, statistical mechanics provided a connection between thermodynamic entropy and the logarithm of the number of microstates in the system’s macrostate. This work is attributed to Ludwig Boltzmann and the Boltzmann entropy [21], *S*, was defined as
(1)$$S=k_{B}\cdot ln\left(W\right)$$
where $k_{B}$ is the thermodynamic unit of the measurement of the entropy and is the Boltzmann constant, *W* is called the thermodynamic probability or statistical weight and is the total number of microscopic states or complexions compatible with the macroscopic state of the system. The Boltzmann entropy, also known as Boltzmann-Gibbs entropy [22] or thermodynamic entropy [23], is a function on phase space and is thus defined for an individual system. Equation (Equation 1) was originally formulated by Boltzmann between 1872 and 1875 [24] but later put into its current form by Max Planck in about 1900 [25].

In 1902, Gibbs published *Elementary Principles in Statistical Mechanics* [26], where he introduced Gibbs entropy. In contrast to Boltzmann entropy, the Gibbs entropy of a macroscopic classical system is a function of a probability distribution over phase space, i.e., an ensemble. The Gibbs entropy [27] in the phase space of system *X*, at time *t*, is defined by the quantity, $S_{G}\left(t\right)$:(2)$$S_{G}\left(t\right)=-k_{B}\int_{X}f_{t}\left(x\right)\left[logf_{t}\left(x\right)-1\right]dx$$
where $k_{B}$ is the Boltzmann’s constant and $f_{t}$ is the density of the temporal evolution of the probability distribution in the phase space.

Boltzmann entropy was defined for a macroscopic state of a system, while Gibbs entropy is the generalization of the Boltzmann entropy for over an ensemble, which is over the probability distribution of macrostates [21,28].

In 1928, the electrical engineer Ralph Hartley [29] proposed that the amount of information associated with any finite set of entities could be understood as a function of the set’s size. Hartley defined the amount of information h(X) associated with the finite set *X* as the logarithm to some base *b* of the $X^{\prime }s$ size, as shown in Equation (Equation 3). This amount is known as Hartley entropy and is related to the particular cases of Rényi entropy (see Section 2.4.1).
(3)$$h\left(X\right)=log_{b}\left|X\right|$$

In 1932, Von Neumann generalized Gibbs entropy to quantum mechanics, and it is known as Von Neumann entropy (quantum Gibbs entropy or quantum entropy) [30]. The quantum entropy was initially defined as Shannon entropy associated with the density matrix’s eigenvalues, but sixteen years before Shannon entropy was defined [31]. Von Neumann entropy is the quantum generalization of Shannon entropy [32].

In 1948, the American Claude Shannon published “*A mathematical theory of communication*” in July and October issues of Bell System technical journal [33]. He proposed the notion of entropy to measure how the information within a signal can be quantified with absolute precision as the amount of unexpected data contained in the message. This measure is known as Shannon entropy (SE) and is defined as:(4)$$SE\left(X\right)=-\sum_{i}p_{i}\cdot ln\left(p_{i}\right)$$
where $X=\{x_{i},i=1,\ldots ,N\}$ is a time-series, 0ln0=0 by convention and $p_{i}$ represents the probability of $x_{i}$, i=1,…,N. Therefore $p_{i}>0$∀i and $\sum p_{i}=1$.

In 1949, at the request of Shannon employer, Warren Weaver, the paper of Shannon is republished as a book [34], preceded by an introductory exhibition by Weaver. Weaver’s text [35] attempts to explain how Shannon’s ideas can extend far beyond their initial objectives to all sciences that address communication problems in the broad sense. Weaver is sometimes cited as the first author, if not the only author of information theory [36]. Nevertheless, as Weaver himself stated: “*No one could realize more keenly than I do that my own contribution to this book is infinitesimal as compared with Shannon’s*” [37]. Shannon and Weaver, in the book “*The Mathematical Theory of Communication*” [38], referred to Tolman (1938), who in turn attributes to Pauli (1933), the definition of entropy Shannon used [39].

Many authors [7,40,41] used the term Boltzmann-Gibbs-Shannon entropy (BGS), generalizing the entropy expressions of Boltzmann, Gibbs, and Shannon and following the ideas of Stratonovich [42] in 1955:(5)$$S_{BGS}\left(p_{1},\ldots ,p_{w}\right)=-k_{B}\sum_{i=1}^{W}p_{i}\cdot ln\left(p_{i}\right)$$
where $k_{B}$ is the Boltzmann constant, *W* is the number of microstates consistent with the macroscopic constraints of a given thermodynamical system and $p_{i}$ is the probability that the system is in the microstate *i*.

In 1957, Khinchin [43] proposed an axiomatic definition of the Boltzmann entropy, based on four requirements, known as the Shannon-Khinchin axioms [8]. He also demonstrated that Shannon entropy is generalized by
(6)$$H\left(X\right)=-k\sum_{i}p_{i}\cdot log\left(p_{i}\right)$$
where *k* is a positive constant representing the desired unit of measurement. This property enables us to change the logarithm base in the definition, i.e., $SE_{b}\left(X\right)=log_{b}SE_{a}\left(X\right)$. Entropy can be changed from one base to another by multiplying by the appropriate factor [44]. Therefore, depending on the application area, instead of the natural logarithm in Equation (Equation 4), one can use logarithms from other bases.

There are different approaches to the derivation of Shannon entropy based on different postulates or axioms [44,45,46,47].

Figure 2 represents the origin of the universe of entropies that we propose in this paper, as well as the relationships between the entropies that we describe in this section.

### 2.2. Entropies Derived from Shannon Entropy

Based on Shannon entropy, many researchers have been devoted to enhancing the performance of Shannon entropy for more accurate complexity estimation, such as differential entropy, spectral entropy, tone-entropy, wavelet entropy, empirical mode decomposition energy entropy, and Δ−entropy. Throughout this paper, other entropies related to Shannon entropy have been analyzed, such as Rényi, Tsallis, permutation, and dispersion entropies. However, due to its context, we decided to include them in other sections.

In Figure 3, we represent how the entropies described below are related to each other.

#### 2.2.1. Differential Entropy

Shannon entropy was formalized for discrete probability distributions. However, the concept of entropy can be extended to continuous distributions through a quantity known as differential entropy (DE) (also referred to as continuous entropy). The concept of DE was introduced by Shannon [33] like a continuous-case extension of Shannon entropy [44]. However, further analysis reveals several shortcomings that render it far less useful than it appears.

The information-theoretic quantity for continuous one-dimensional random variables is differential entropy. The DE for a continuous time-series *X* with probability density p(x) is defined as:(7)$$DE\left(X\right)=-\int_{S}p_{i}\cdot ln\left(p_{i}\right)$$
where *S* is the support set of the random variable.

In the discrete case, we had a set of axioms from which we derived Shannon entropy and, therefore, a collection of properties that the measure must present. Charles Marsh [48] stated that “*continuous entropy on its own proved problematic*,” the differential entropy is not a limit of the Shannon entropy for n→∞. On the contrary, it differs from the limit of the Shannon entropy by an infinite displacement [48]. Among other problems, continuous entropy can be negative, while discrete entropy is always non-negative. For example, when the continuous random variable *U* is uniformly distributed over the interval (a,b), p(u)=1/(b−a) Equation (Equation 7) results in:(8)DE(U)=ln(b−a).

The entropy value obtained in Equation (Equation 8) is negative when the length of the interval is <1.

#### 2.2.2. Spectral Entropy

In 1992, Kapur et al. [49] proposed spectral entropy (SpEn) that uses the power spectral density $\hat{P}\left(f\right)$ obtained from the Fourier transformation method [50]. The power spectral density represents the distribution of power as a function of frequency. Thus, the normalization of $\hat{P}\left(f\right)$ yields a probability density function. Using the definition of Shannon entropy, SpEn can be defined as:(9)$$SpEn=\sum_{i=f_{l}}^{f_{h}}p_{i}\cdot log\left(p_{i}\right)$$
where $\left[f_{l},f_{h}\right]$ is the frequency band. Spectral entropy is usually normalized $SpEn/logN_{f}$, where $N_{f}$ is the number of frequency components in the range $\left[f_{l},f_{h}\right]$.

#### 2.2.3. Tone-Entropy

In 1997, Oida et al. [51] proposed the tone-entropy (T-E) analysis to characterize the time-series of percentage index (PI) of heart period variation. In this paper, the authors used the tone, and the entropy to characterize the time-series PI. Considered a time-series $X=\{x_{i},i=1,\ldots ,N\}$, PI is defined as:(10)$$PI\left(i\right)=\frac{\left(x_{i}-x_{i+1}\right)100}{x_{i}}.$$

The Tone is defined as the first-order moment (arithmetic average) of this PI time-series as:(11)$$Tone=\frac{1}{N-1}\sum_{i=1}^{N}PI\left(i\right).$$

Entropy is defined from the PI’s probability distribution using Shannon’s Equation (Equation 4) with $log_{2}$ instead of ln.

#### 2.2.4. Wavelet Entropy

Wavelet entropy (WaEn) was introduced by Rosso and Blanco [52] in 2001. The WaEn is an indicator of the disorder degree associated with the multi-frequency signal response [11]. The definition of WaEn is given as follows:(12)$$WaEn=-\sum_{i<0}p_{i}\cdot log\left(p_{i}\right)$$
where $p_{i}$ denotes the probability distribution of time-series, and *i* represents different resolution levels.

#### 2.2.5. Empirical Mode Decomposition Energy Entropy

In 2006, Yu et al. [53] proposed the Empirical Mode Decomposition Energy entropy (EMDEnergyEn) that quantifies the regularity of time-series with the help of the intrinsic mode functions (IMFs) [12] obtained by the empirical mode decomposition (EMD).

Assume that we have obtained *n* IMFs, three steps are required to obtain the EMDEnergyEn as follows:Calculate $E_{i}$ energy for each *i*th IMFs $c_{i}$:
(13)$$E_{i}=\sum_{j=1}^{m}{\left|c_{i}\left(j\right)\right|}^{2}$$
where *m* represents the length of IMF.Calculate the total energy of these *n* efficient IMFs:
(14)$$E=\sum_{i=1}^{n}E_{i}.$$Calculate the energy entropy of IMF:
(15)$$H_{en}=-\sum_{j=1}^{n}p_{i}\cdot log\left(p_{i}\right)$$
where $H_{en}$ denotes the EMDEnergyEn in the whole of the original signal, and $p_{i}=\frac{Ei}{E}$ denotes the percentage of the energy of the IMF number *i* relative to the total energy entropy.

#### 2.2.6. Δ−Entropy

In 2011, the Δ−entropy was introduced by Chen et al. [54]. This measure is sensitive to the dynamic range of the time-series. The Δ−entropy contains two terms, in which the first term measures the probabilistic uncertainty obtained with Shannon entropy. The second term measures the dispersion in the error variable:(16)$$H_{\Delta }\left(X\right)=-\sum_{i}p_{i}\cdot log\left(p_{i}\right)+log\Delta \left(X\right)$$
where $X=\{x_{i},i=1,\ldots ,N\}$ is the time-series and $\Delta \left(X\right)=\frac{1}{N-1}\sum_{i=1}^{N-1}\left|x_{i+1}-x_{i}\right|$.

The Δ−entropy converges to DE when the scale (Δ(X)) tends to zero. When the scale defaults to the natural numbers (Δ(X)=1 and therefore log(Δ(X))=0) the Δ−entropy is indistinguishable from SE.

### 2.3. Kolmogorov, Topological and Geometric Entropies

In 1958, Kolmogorov [55] introduced the concept of entropy in the dynamical system as a measure-preserving transformation and studied the attendant property of completely positive entropy (*K*-property). In 1959, his student Sinai [56,57] formulated Kolmogorov-Sinai entropy (or metric entropy or Kolmogorov entropy [58]) that is suitable for arbitrary automorphisms of Lebesgue spaces. The Kolmogorov-Sinai entropy is equivalent to a generalized version of SE under certain plausible assumptions [59].

For the case of a state space, it can be partitioned into hypercubes of content $\epsilon^{m}$ in an *m*-dimensional dynamical system and observed at time intervals δ, defining the Kolmogorov-Sinai entropy as [60]: (17)$$H_{KS}=-\lim_{\delta \to 0}\lim_{\epsilon \to 0}\lim_{n\to \infty }\frac{1}{n\delta }\sum_{K_{1},\ldots ,K_{n}}p\left(K_{1},\ldots ,K_{n}\right)\cdot ln\left(p(K_{1},\ldots ,K_{n})\right)=\lim_{\delta \to 0}\lim_{\epsilon \to 0}\lim_{n\to \infty }\frac{1}{n\delta }H_{n}$$
where $p(K_{1},\ldots ,K_{n})$ denote the joint probability that the state of the system is in the hypercube $k_{1}$ at the time t=δ, $k_{2}$ at t=2δ, *…*, in the hypercube $k_{n}$ at t=nδ and $H_{n}=H\left(X_{1},\ldots ,X_{n}\right)=\sum_{i=1}^{n}H\left(X_{i}|X_{i-1},\ldots ,X_{1}\right)$. For stationary processes it can be shown that
(18)$$H_{KS}=-\lim_{\delta \to 0}\lim_{\epsilon \to 0}\lim_{n\to \infty }\left(H_{n+1}-H_{n}\right).$$

It is impossible to calculate Equation (Equation 17) for n→∞ because different estimation methods have been proposed, such as approximate entropy [61] and Rényi entropy [62], and compression [63].

The Kolmogorov-Sinai entropy and conditional entropy coincide for a stochastic process *X*, where $X_{n}$ is the random variable obtained by sampling the process *X* at the present time *n*, and $X_{n-1}=\left\{X_{1},\ldots ,X_{n-1}\right\}$ [64].

In this case, conditional entropy quantifies the amount of information in the current process that cannot be explained by its history. If the process is random, the system produces information at the maximum rate, producing the maximum conditional entropy. If, on the contrary, the process is completely predictable, the system does not produce new information, and conditional entropy is zero. When the process is stationary, the system produces new information at a constant rate, meaning that the conditional entropy does not change over time [65]. Note that conditional entropy is, more broadly, the entropy of a random variable conditioned to the knowledge of another random variable [44].

In 1965, inspired by Kolmogorov-Sinai entropy, the concept of the topological entropy (TopEn) was introduced by Adler et al. [66] to describe the complexity of a single map acting on a compact metric space. The notion of TopEn is similar to the notion of metric entropy: instead of a probability measure space, we have a metric space [67].

After 1965, many researchers proposed other notions of TopEn [68,69,70,71,72]. Most of the new notions extended the concept to more general functions or spaces, but the idea of measuring the complexity of the systems was preserved among all these new notions. In [56], the authors review the notions of topological entropy, give an overview of the relation between the notions and fundamental properties of topological entropy.

In 2018, Rong and Shang [10] introduced TopEn based on time-series to characterizes the total exponential complexity of a quantified system with a single number. The authors began by choosing a symbolic method to transform the time-series $X=\{x_{i},i=1,\ldots ,N\}$ into a symbolic sequence $Y=\{y_{i},i=1,\ldots ,N\}$. At the same time, consider *k* the number of different alphabets of *Y* and P(n) represents the number of different sets of words with length *n*. Note that P(N)=1 and the TopEn of the time-series was defined as:(19)$$h_{n}=\frac{log_{k}P\left(n\right)}{n}\;\left(1\leq n\leq N\right).$$

The maximum of P(n) is $k^{n}$, then TopEn can reach a maximum of 1, while TopEn’s minimum value is 0, which is reached when n=N. Addabbo and Blackmore [73] showed that metric entropy, with the imposition of some additional properties, is a special case of topological entropy and Shannon entropy is a particular form of metric entropy.

In 1988, Walczak et al. [74] introduced the geometric entropy (or foliation entropy) to study a foliation dynamics, which can be considered as a generalization of TopEn of a single group [75,76,77]. Recently, Rong and Shang [10] proposed geometric entropy for time-series based on the multiscale method (see Section 2.10) and the original definition of geometric entropy provided by Walczak. To calculate the value of TopEn and geometric entropy in their work, the authors used horizontal visibility graphs [78] to transform the time-series into a symbolic series. More details about horizontal visibility graphs in Section 2.8.

In Figure 4, we show the relationships between the entropies described in this section.

### 2.4. Rényi Entropy

In 1961, Rényi entropy (RE) or q-entropy was introduced by Alfréd Rényi [79] and played a significant role in information theory. Rényi entropy, a generalization of Shannon entropy, is a family of functions of order q ($R_{q}$) for quantifying the diversity, uncertainty, or randomness of a given system defined as:(20)$$R_{q}\left(X\right)=\frac{1}{1-q}log_{2}\sum_{i}{p_{i}}^{q}.$$

In the following sections, we describe particular cases of Rényi entropy, ϵ−smooth Rényi entropy, Rényi entropy for a continuous random variable, and discuss issues with the different definitions of quadratic entropy.

Figure 5 illustrates the relations between Shannon entropy, Rényi entropy, particular cases of Rényi entropy, and ϵ−smooth Rényi entropy.

#### 2.4.1. Particular Cases of Rényi Entropy

There are some particular cases of Rényi entropy [80,81,82], for example, if q=0, $R_{0}=\frac{1}{1-0}log_{2}\sum_{i=1}^{N}{p_{i}}^{0}=log_{2}N$ is the Hartley entropy (see Equation (Equation 3)).

The Rényi entropy converges to the well-known Shannon entropy, eventually, with multiplication by a constant resulting from the base change property, i.e., when q→1 and $log_{2}$ is replaced by ln.

When q=2, $R_{2}=-log_{2}\sum_{i=1}^{N}{p_{i}}^{2}$ is called collision entropy [80] and is the negative logarithm of the likelihood of two independent random variables with the same probability distribution to have the same value. The collision entropy measures the probability for two elements drawn according to this distribution to collide. Many papers refers to Rényi entropy [81] when using q=2, even when that choice it is not explicit.

The concept of maximum entropy (or Max-Entropy) arose in statistical mechanics in the nineteenth century and has been advocated for use in a broader context by Edwin Thompson Jaynes [83] in 1957. The Max-Entropy can be obtained from Rényi entropy when q→−∞, and the limit exists, as:(21)$$R_{-\infty }=-log_{2}\left(min_{i=1,\ldots ,N}p_{i}\right),$$

It is the largest value of $R_{q}$, which justifies the name maximum entropy. The particular cases of Rényi entropy when q→∞, and the limit exists as
(22)$$R_{\infty }=-log_{2}\left(max_{i=1,\ldots ,N}p_{i}\right)$$
is called minimum entropy (or Min-Entropy) because it is the smallest value of $R_{q}$. The Min-Entropy was proposed by Edward Posner [84] in 1975. According to reference [85], the collision entropy and Min-Entropy are related by Equation by the following:(23)$$R_{\infty }\leq R_{2}\leq 2R_{\infty }.$$

#### 2.4.2. ϵ−Smooth Rényi Entropy

In 2004, Renner and Wolf [86] proposed the ϵ-smooth Rényi entropy for characterizing the fundamental properties of a random variable, such as the amount of uniform randomness that may be extracted from the random variable:(24)$$R_{q}^{\epsilon }\left(X\right)=\frac{1}{1-q}\underset{Q\in B^{\epsilon }\left(P\right)}{\inf}log_{2}\left(\sum_{z\in Z}Q{\left(z\right)}^{q}\right)$$
where $B^{\epsilon }\left(P\right)=\{Q:(\sum_{z}\left|P\left(z\right)-Q\left(z\right)|)/2\leq \epsilon \right\}$ is the set of probability distributions which are ϵ-close to *P*; *P* is the probability distribution with range *Z*; q∈[0,∞] and ϵ≥0. For the particular case of a significant number of independent and identically distributed (i.i.d.) random variables, ϵ−smooth Rényi entropy approaches the Rényi entropy. In this special case, if q→1 the ϵ−smooth Rényi entropy approaches the Rényi entropy.

#### 2.4.3. Rényi entropy for Continuous Random Variables and the Different Definition of Quadratic Entropy

According to Lake [87], if *X* is an absolutely continuous random variable with density *f*, the Rényi entropy of order *q* (or *q*-entropy) is defined as:(25)$$R_{q}\left(X\right)=\frac{1}{1-q}log_{2}\left(\int_{-\infty }^{\infty }f{\left(x\right)}^{q}\right)$$
where letting *q* tend to 1, and using L’Hospitals rule results in differential entropy, i.e., $DE\left(x\right)=R_{1}\left(X\right)$. Lake [87] uses the term quadratic entropy when q=2.

In 1982, Rao [88,89] gave a different definition for quadratic entropy. He introduced quadratic entropy as a new measure of diversity in biological populations, which considers differences between categories of species. For the discrete and finite case, the quadratic entropy is defined as:(26)$$\sum_{i=1}^{S}\sum_{j=1}^{S}d_{i,j}p_{i}p_{j}$$
where $d_{i,j}$ is the difference between the *i*-th and the *j*-th category and $p_{1},\ldots ,p_{s}$ ($\sum p_{i}=1$) are the theoretical probabilities corresponding to the *s* species in the multinomial model.

### 2.5. Havrda–Charvát Structural α−Entropy and Tsallis Entropy

The Havrda–Charvát Structural α−entropy was introduced in 1967 within information theory [90]. It may be considered as a new generalization of the Shannon entropy, different from the generalization given by Rényi. The Havrda–Charvát Structural α−entropy, $S_{\alpha }$, was defined as:(27)$$S_{\alpha }=\frac{2^{\alpha -1}}{2^{\alpha -1}-1}\left(1-\sum_{i}p_{i}^{\alpha }\right)$$
for α>0 and α≠1. When α→1, the Havrda–Charvát Structural α-entropy converges for Shannon entropy minus multiplication by a constant (1/ln2).

In 1988, the concept of Tsallis entropy (TE) was introduced by Constantino Tsallis as a possible generalization of Boltzmann-Gibbs entropy to nonextensive physical systems and prove that the Boltzmann-Gibbs entropy is recovered as q→1 [7,91]. TE is identical in form to Havrda–Charvát Structural α−entropy and Constantino Tsallis defined it as:(28)$$S_{q}=\frac{k}{q-1}\left(1-\sum_{i}p_{i}^{q}\right),\;q\in R$$
where *k* is a positive constant [91]. In particular, when q→1 implies that
(29)$$S_{1}=\lim_{q\to 1}S_{q}=-k\sum_{i}p_{i}\cdot ln\left(p_{i}\right)$$

When k=1 in Equation (Equation 29) we recover Shannon entropy [92]. In Equation (Equation 29) if we consider *k* to be the Boltzmann constant and $p_{i}=\frac{1}{N}$, ∀i, we recover the Boltzmann entropy to W=N.

Figure 6 summarizes the relationships between the entropies that we describe in this section.

### 2.6. Permutation Entropy and Related Entropies

Bandt and Pompe [13] introduced permutation entropy (PE) in 2002. PE is an ordinal analysis method, in which a given time-series is divided into a series of ordinal patterns for describing the order relations between the present and a fixed number of equidistant past values [93].

A time-series, as a set of *N* consecutive data points, is defined as $X=\{x_{i},i=1,\ldots ,N\}$. PE can be obtained by the following steps. From the original time-series *X*, let us define the vectors $X_{m}^{\tau }\left(i\right)$ as:(30)$$X_{i}=X_{m}^{\tau }\left(i\right)=\left(x_{i},x_{i+\tau },x_{i+2\tau },\ldots ,x_{i+(m-1)*\tau }\right)$$
with i=1,…,N−(m−1)∗τ, where *m* is the embedding dimension, and τ is the embedding lag or time delay. Reconstruct the phase space so that the time-series maps to a trajectory matrix $X_{m}^{\tau }$,
(31)$$X_{m}^{\tau }={\left(X_{1},X_{2},\ldots ,X_{N-(m-1)*\tau }\right)}^{\Gamma }.$$

Each state vector $X_{i}$ has an ordinal pattern. It is determined by a comparison of neighboring values. Sort $X_{i}$ in ascending order, and the ordinal pattern $\pi_{i}$ is the ranking of $X_{i}$. The trajectory matrix $X_{m}^{\tau }$ is thus transformed into ordinal pattern matrix,
(32)$$\pi ={\left(\pi_{1},\pi_{2},\ldots ,\pi_{N-(m-1)*\tau }\right)}^{\Gamma }$$

It is called permutation because there is a transformation from $X_{i}$ to $\pi_{i}$ and for a given embedding dimension *m*, at most m! permutations exist in total.

Select all distinguishable permutations and number them $\pi_{j}$, j=1,2,…,m!. For all the m! possible permutations $\pi_{j}$, the relative frequency is calculated by
(33)$$p\left(\pi_{j}\right)=\frac{\#\left\{X_{i}|X_{i}\;\mathrm{has}\;\mathrm{ordinal}\;\mathrm{pattern}\;\pi_{j}\right\}}{N-(m-1)\tau }$$
where 1≤i≤N−(m−1)τ and # represents the number of elements.

According to information theory, the information contained in $\pi_{j}$ is measured as $-ln\left(p(\pi_{j})\right)$ and PE is finally computed as:(34)$$H\left(m\right)=-\sum_{j=1}^{m!}p\left(\pi_{j}\right)\cdot ln\left(p(\pi_{j})\right).$$

Since H(m) can maximally reach ln(m!), PE is generally normalized as:(35)$$PE\left(m\right)=-\frac{\sum_{j=1}^{m!}p\left(\pi_{j}\right)\cdot ln\left(p(\pi_{j})\right)}{ln(m!)}.$$

This is the information contained in comparing *m* consecutive values of the time-series. Bandt and Pompe [13] defined the permutation entropy per symbol of order *m* (pe(m)), dividing by m−1 since comparisons start with the second value:(36)$$pe\left(m\right)=\frac{PE(m)}{m-1}$$

In reference [13], “*Permutation entropy — a natural complexity measure for time-series*” the authors also proposed the sorting entropy (SortEn) defined as: $d_{m}=H\left(m\right)-H\left(m-1\right)$, $d_{2}=H\left(2\right)$. This entropy determines the information contained in sorting the *m*th value among the previous m−1 when their order is already known.

Based on the probability of the permutation $p_{j}=p\left(\pi_{j}\right)$, other entropy can be defined, e.g., Zhao et al. [14], in 2013, introduced Rényi permutation entropy (RPE) as:(37)$$RPE_{q}=\frac{1}{1-q}log_{2}\sum_{j=1}^{m!}{p_{j}}^{q}.$$

Zunino et al. [92], based on the definition of TE, proposed the normalized Tsallis permutation entropy (NTPE) and defined it as:(38)$$NTPE_{q}=\frac{\sum_{j=1}^{m!}\left(p_{j}-p_{j}^{q}\right)}{1-m{!}^{1-q}}.$$

In Figure 7, we showed how Tsallis, Rényi, Shannon and the entropies described in this section are related to each other. The rank-based and bubble entropies, described the Section 2.7, are related to permutation entropy and are also represented in the Figure 7.

### 2.7. Rank-Based Entropy and Bubble Entropy

Citi et al. [94] proposed the rank-based entropy (RbE), in 2014. The RbE consists of an alternative entropy metric based on the amount of shuffling required for ordering the mutual distances between *m*-long vectors when the next observation is considered, that is, when the corresponding m+1-long vectors are considered [94]. Operationally, RbE can be defined by the following steps:Compute, for 1≤i<j≤N−m, the mutual distances vectors: $d_{k(i,j)}={∥v_{m,i}-v_{m,j}∥}_{\infty }$ and $d_{k(i,j)}^{{}^{\prime }}=\left|x_{i+m}-x_{j+m}\right|$ where ${∥v_{m,i}∥}_{\infty }$ is the infinity norm of vector $v_{m,i}=\left\{x_{i},x_{i+1},\ldots ,x_{i+m-1}\right\}$ that is, ${∥v_{m,i}∥}_{\infty }=max_{1\leq l\leq i+m-1}\left|x_{l}\right|$ and k=k(i,j) is the index assigned for each (i,j) pair, with 1≤k≤K=(N−m−1)(N−m)/2.Consider vector $d_{k}$ and find the permutation π(k) such that the vector $S_{k}=d_{\pi (k)}$ is sorted in ascending order. Now, if the system is deterministic, we expect that if the vectors $v_{m,i}$ and $v_{m,j}$ are close, then the new observation from each vector $x_{i+m},x_{j+m}$ should be close too. In other words, $S_{k}^{{}^{\prime }}=d_{\pi (k)}^{{}^{\prime }}$ should be almost sorted too. Compute inversion count which is a measure of a vector‘s disorder.Determine the largest index $k_{\rho }$ satisfying $S_{k_{\rho }}<\rho$ and compute the number *I* of inversion pairs $\left(k_{1},k_{2}\right)$ such that $k_{1}<k_{\rho }$, $k_{1}<k_{2}$ and $S_{k_{1}}^{{}^{\prime }}>S_{k_{2}}^{{}^{\prime }}$.Compute the RbE as:
(39)$$RbE=-ln\left(1-\frac{I}{\left(2K-k_{\rho }-1\right)k_{\rho }/2}\right).$$

The concept of bubble entropy (BEn) was introduced by Manis et al. [95] in 2017 and it is based on permutation entropy (see Section 2.6), where the vectors in the embedding space are ranked and this rank is inspired by rank-based entropy.

The computation of BEn is as follows:Sort each vector $X_{i}$, defined in Equation (Equation 30), of m−1 elements in ascending order, counting the number of swaps $n_{i}$ necessary. The $n_{i}$ number is obtained by bubble sort [96]. For more details about bubble sort see paper [97].Compute an histogram from $n_{i}$ values and normalize it by N−m, to obtain the probabilities $p_{i}$ (describing how likely a given number of swaps $n_{i}$ is).Consider $RPE_{2}^{m-1}$ according to Equation (Equation 37) for q=2 and from $p_{i}$ compute the $RPE_{swaps}^{m-1}$ [98]:
(40)$$RPE_{swaps}^{m-1}=-log_{2}\sum_{i=1}^{n}{p_{i}}^{2}.$$Repeat steps 1 to 3 to compute $RPE_{swaps}^{m}$.Compute BEn by:
(41)$$BEn=\frac{\left(RPE_{swaps}^{m}-RPE_{swaps}^{m-1}\right)}{log_{2}\left(\frac{m}{m-2}\right)}.$$

### 2.8. Topological Information Content, Graph Entropy and Horizontal Visibility Graph Entropy

In the literature, there are variations in the definition of graph entropy [99]. Inspired by Shannon’s entropy, the first researchers to define and investigate the entropy of graphs were Rashevsky [100], Trucco [101], and Mowshowitz [102,103,104,105]. In 1955, Rashevsky introduced the information measures for graphs G=(V,E) called topological information content and defined as:(42)$$V_{I}\left(G\right)=-\sum_{i=1}^{k}\frac{|N_{i}|}{\left|V\right|}log\left(\frac{|N_{i}|}{\left|V\right|}\right)$$
where $|N_{i}|$ denotes the number of topologically equivalent vertices in the *i*th vertex orbit of *G* and *k* is the number of different orbits. Vertices are considered as topologically equivalent if they belong to the same orbit of a graph *G*. In 1956, Trucco [101] introduced similar entropy measures applying the same principle to the edge automorphism group.

In 1968, Mowshowitz [102,103,104,105] introduced an information measure using chromatic decompositions of graphs and explored the properties of structural information measures relative to two different equivalence relations defined on the vertices of a graph. Mowshowitz [102] defined graph entropy as:(43)$$I\left(G\right)=min_{\hat{V}}\left\{-\sum_{i=1}^{h}\frac{n_{i}\left(\hat{V}\right)}{\left|V\right|}log\left(\frac{n_{i}\left(\hat{V}\right)}{\left|V\right|}\right)\right\}$$
where $\hat{V}=\left\{V_{i}|1\leq i\leq h\right\}$, $|V_{i}|=n_{i}\left(V\right)$ denotes an arbitrary chromatic decomposition of a graph G, h=χ(G) is the chromatic number of *G*.

In 1973, János Körne [106] introduced a different definition of graph entropy linked to problems in information and coding theory. The context was to determine the performance of the best possible encoding of messages emitted by an information source where the symbols belong to a finite vertex set *V*. According to Körne, the graph entropy of *G*, denoted by H(G) is defined as:(44)$$H\left(G\right)=\min_{X,Y}I\left(X;Y\right)$$
where *X* is chosen uniformly from *V*, *Y* ranges over independent sets of *G*, the joint distribution of *X* and *Y* is such that X∈Y with probability one, and I(X;Y) is the mutual information of *X* and *Y*. We will not go deeper into the analysis of the graph entropy defined by Körner because it involves the concept of mutual information.

There is no unanimity in the research community regarding the first author that defined graph entropy. Some researchers consider that graph entropy was defined by Mowshowitz in 1968 [50,99] but other researchers consider that it was introduced by János Körne in 1973 [107,108]. New entropies have emerged based on both concepts of graph entropy. When the term graph entropy is mentioned, care must be taken and understand the underlying definition.

The concept of horizontal visibility graph entropy, a measure based on one definition of graph entropy and in the concept of horizontal visibility graph, was proposed by Zhu et al. [109], in 2014. The horizontal visibility graph is a type of complex network, based on a visibility algorithm [78], and was introduced by Luque et al. [110], in 2009. Let a time-series $X=\{x_{i},i=1,\ldots ,N\}$ be mapped into a graph G(V,E) where a time point $x_{i}$ is mapped into a node $v_{i}$. The relationship between any two points $\left(x_{i},x_{j}\right)$ is represented by an edge $e_{ij}$, which are connected if and only if the maximal values between $x_{i}$ and $x_{j}$ are less than both of them. Therefore, each edge can be defined as [111]:$$e_{i,j}=\left\{\begin{array}{cc} 1 & \;x_{j}>max\left(x\left[\left(j+1\right)\ldots \left(i-1\right)\right]\right)\\ 1 & \;j+1=i\\ 0 & \mathrm{other}\;\mathrm{cases} \end{array}\right.$$
where $e_{ij}=0$ implies that the edge does not exist, otherwise it does.

The graph entropy, based on either vertices or edges, can be computed as follows:(45)$$GE=-\sum_{i=1}^{n}p\left(k\right)\cdot log\left(p(k)\right)$$
where p(k) is a probability degree *k* over a degree sequence of the graph *G*. It is obtained by counting the number of nodes having degree *k* divided by the size of the degree sequence. Notice that, the more fluctuating the degree sequence is, the larger the graph entropy is. There are others graph entropy calculation methods based on either vertices or edges [99].

We summarize how the entropies described in this section are related to each other, in Figure 8.

### 2.9. Approximate and Sample Entropies

Pincus [112] introduced the approximate entropy (ApEn), in 1991. ApEn is derived from Kolmogorov entropy [113] and it is a technique used to quantify the amount of regularity and the unpredictability of fluctuations over time-series data [114]. In order to calculate the ApEn the new series of a vector of length *m* (embedding dimension) are constructed, $X_{m}^{1}$, based on Equation (Equation 30). For each vector $X_{m}^{1}\left(i\right)$, the value $C_{m}^{r}\left(i\right)$, where *r* is referred as a tolerance value, is computed as:(46)$$C_{m}^{r}\left(i\right)=\frac{\mathrm{number}\;\mathrm{of}\;d[X_{m}^{1}\left(i\right),X_{m}^{1}\left(j\right)]\leq r}{N-m+1},\;\forall j.$$

Here the distance between the vector $X_{m}^{1}\left(i\right)$ and its neighbor $X_{m}^{1}\left(j\right)$ is:(47)$$d\left[X_{m}^{1}\left(i\right),X_{m}^{1}\left(j\right)\right]=max_{k=1,\ldots ,m}\left|x\left(i+k-1\right)-x\left(j+k-1\right)\right|.$$

The value $C_{m}^{r}\left(i\right)$ can also be defined as:(48)$$C_{m}^{r}\left(i\right)=\frac{1}{N-m+1}\sum_{j=1}^{N-m+1}\theta \left(max_{k=1,\ldots ,m}\left|x\left(i+k-1\right)-x\left(j+k-1\right)\right|-r\right)$$
where θ is the Heaviside function
$$\theta \left(t\right)=\left\{\begin{array}{cc} 1 & \;\mathrm{if}\;t\leq 0\\ 0 & \;\mathrm{if}\;t>0 \end{array}\right.$$

Next, the average of the natural logarithm of $C_{m}^{r}\left(i\right)$ is computed for all *i*:(49)$$\Phi_{m}^{r}=\frac{1}{N-m+1}\sum_{i=1}^{N-m+1}ln\left(C_{m}^{r}\left(i\right)\right).$$

Since in practice *N* is a finite number, the statistical estimate is computed as:$$ApEn\left(m,r\right)=\left\{\begin{array}{cc} \Phi_{m}^{r}-\Phi_{m+1}^{r} & \;\mathrm{for}\;m>0\\ -\Phi_{1}^{r} & \;\mathrm{for}\;m=0 \end{array}\right.$$

The disadvantages of ApEn are: it lacks relative consistency, its strong dependence on data length and is often lower than expected for short records [15]. To overcome the disadvantages of ApEn, in 2000 Richman and Moorman [15] proposed the sample entropy (SampEn) to replace ApEn by excluding self-matches and thereby reducing the computing time by one-half in comparison with ApEn. For the SampEn [15] calculation, the same parameters defined for the ApEn, *m*, and *r* are required. Considering *A* as the number of vector pairs of length m+1 having $d[X_{m}^{1}\left(i\right),X_{m}^{1}\left(j\right)]\leq r$, with i≠j and *B* as the total number of template matches of length *m* also with i≠j, the SampEn is defined as:(50)$$SampEn=-ln\frac{A}{B}.$$

Many entropies related to ApEn and SampEn have been created. In Figure 9, some entropies related to ApEn and SampEn are represented, which are described in the following sections.

#### 2.9.1. Quadratic Sample Entropy, Coefficient of Sample Entropy and Intrinsic Mode Entropy

The SampEn has a strong dependency on the size of the tolerance *r*. Normally, smaller *r* values lead to higher and less confident entropy estimates because of falling numbers of matches of length *m* and, to an even greater extent, matches of length m+1 [16].

In 2005, Lake [87] introduced the concept of quadratic sample entropy (QSE) (Lake called it quadratic differential entropy rate) to solve the aforementioned problem. QSE normalizes the value of *r* and allows any *r* for any time-series and the results to be compared with any other estimate. The QSE is defined as follows:(51)$$QSE=-ln\left(\frac{\frac{A}{B}}{2r}\right)=SampEn+ln\left(2r\right).$$
where *A* is the number of vector pairs of length m+1 having $d[X_{m}^{1}\left(i\right),X_{m}^{1}\left(j\right)]\leq r$, with i≠j and *B* is the total number of template matches of length *m* also with i≠j, as in SampEn calculation.

In 2010, derived from QSE it was introduced the coefficient of sample entropy (COSEn) [16]. This measure was first devised and applied to the detection of atrial fibrillation through the heart rate. CosEn is computed similarly as QSE:(52)$$COSEn=-ln\left(\frac{\frac{A}{B}}{2r}\right)=SampEn-ln\left(2r\right)-ln\left(\mu \right)$$
where μ is the mean value of the time-series.

In 2007, Amoud et al. [115] introduced the intrinsic mode entropy (IME). The IME is essentially the SampEn computed on the cumulative sums of the IMF [12] obtained by the EMD.

#### 2.9.2. Dispersion Entropy and Fluctuation-Based Dispersion Entropy

The SampEn is not fast enough, especially for long signals, and PE, as a broadly used irregularity indicator, considers only the order of the amplitude values and hence some information regarding the amplitudes may be discarded. In 2016, to solve these problems, Rostaghi and Azami [18] proposed the dispersion entropy (DispEn) applied to a univariate signal $X=x_{1},x_{2},...,x_{N}$ whose algorithm is as follows:First, $x_{j}\left(j=1,2,\ldots ,N\right)$ are mapped to *c* classes, labeled from 1 to *c*. The classified signal is $u_{j}\left(j=1,2,\ldots ,N\right)$. To do so, there are a number of linear and nonlinear mapping techniques. For more details see [116].Each embedding vector $U_{m}^{\tau ,c}\left(i\right)$ with *m* embedding dimension and τ time delay is created according to $U_{m}^{\tau ,c}\left(i\right)=\left(u_{i}^{c},u_{i+\tau }^{c},u_{i+2\tau }^{c},\ldots ,u_{i+(m-1)\tau }^{c}\right)$ with i=1,…,N−(m−1)τ. Each time-series $U_{m}^{\tau ,c}\left(i\right)$ is mapped to a dispersion pattern $\pi_{v_{0}v_{1}\ldots v_{m-1}}$, where $u_{i}^{c}=v_{0}$, $u_{i+\tau }^{c}=v_{1}$, ..., $u_{i+(m-1)\tau }^{c}=v_{m-1}$. The number of possible dispersion patterns that can be assigned to each time-series $U_{m}^{\tau ,c}\left(i\right)$ is equal to $c^{m}$, since the signal has *m* members and each member can be one of the integers from 1 to *c* [18].For each $c^{m}$ potential dispersion patterns $\pi_{v_{0}v_{1}\ldots v_{m-1}}$, their relative frequency is obtained as follows:
(53)$$p\left(\pi_{v_{0}v_{1}\ldots v_{m-1}}\right)=\frac{\#\left\{i|i\leq N-\left(m-1\right)\tau ,\;\mathrm{has}\;\mathrm{type}\;\pi_{v_{0}v_{1}\ldots v_{m-1}}\right\}}{N-(m-1)\tau }$$
where # represents their cardinality.Finally, the DispEn value is calculated, based on the SE definition of entropy, as follows:
(54)$$DispEn\left(X,m,c,\tau \right)=-\sum_{\pi =1}^{c^{m}}p\left(\pi_{v_{0}v_{1}\ldots v_{m-1}}\right)\cdot ln\left(p(\pi_{v_{0}v_{1}\ldots v_{m-1}})\right).$$

In 2018, Azami and Escudero [116] proposed the fluctuation-based dispersion entropy (FDispEn) as a measure to deal with time-series fluctuations. FDispEn considers the differences between adjacent elements of dispersion patterns. According to the authors, this forms vectors with length m−1, which each of their elements changes from −c+1 to c−1, soon, becoming ${\left(2c-1\right)}^{m-1}$ potential fluctuation-based dispersion patterns. The only difference between DispEn and FDispEn algorithms is the potential patterns used in these two approaches [116].

#### 2.9.3. Fuzzy Entropy

The uncertainty resulting from randomness is best described by probability theory, while the aspects of uncertainty resulting from imprecision are best described by the fuzzy sets introduced by Zadeh [117], in 1965. In 1972, De Luca and Termini [118] used the concept of fuzzy sets and introduced a measure of fuzzy entropy that corresponds to Shannon’s probabilistic measure of entropy. Over the years, other concepts of fuzzy entropy have been proposed [119]. In 2007, Chen et al. [120] introduced fuzzy entropy (FuzzyEn), a measure of time-series regularity, for the characterization of surface electromyography signals. In this case, FuzzyEn is the negative natural logarithm of the probability that two similar vectors for *m* points remain similar for the next m+1 points. This measure of FuzzyEn is similar to ApEn and SampEn, replaces the 0-1 judgment of Heaviside function associated with ApEn and SampEn by a fuzzy relationship function [121], the family of exponential functions $exp(-d_{ij}^{n}/r)$, to get a fuzzy measurement of the two vectors similarity based on their shapes. This measure also comprises the removal of the local baseline which may allow for minimizing the effect of non-stationarity in the time-series. Besides possessing the good properties of SampEn superior to ApEn, FuzzyEn also succeeds in giving an entropy definition for the case of small parameters. The method can also be applied to noisy physiological signals with relatively short databases [50]. Consider a time-series $\left\{x_{i},i=1,\ldots ,N\right\}$ with embedding dimension *m* to calculate FuzzyEn form vector sequence as follows:(55)$$X_{m}\left(i\right)=\left\{x_{i},x_{i+1},x_{i+2},\ldots ,x_{i+m-1}\right\}-x_{0}\left(i\right)$$
with i=1,…,N−m+1. In Equation (Equation 55), $X_{m}\left(i\right)$ represents *m* consecutive *x* values, starting with the *i*th point and generalized by removing a baseline:(56)$$x_{0}\left(i\right)=\frac{1}{m}\sum_{j=0}^{m-1}x_{i+j}.$$

For computing FuzzyEn, consider a certain vector $X_{m}\left(i\right)$, define the distance function $d_{m,ij}$ between $X_{m}\left(i\right)$ and $X_{m}\left(j\right)$ as the maximum absolute difference of the corresponding scalar components:(57)$$d_{m,ij}=d\left[X_{m}\left(i\right),X_{m}\left(j\right)\right]=max_{k=0,\ldots ,m-1}\left|x_{i+k}-x_{0}\left(i\right)-x_{j+k}-x_{0}\left(j\right)\right|$$
with i,j=1,…,N−m, j≠i.

Given *n* and *r*, calculate the similarity degree $D_{m,ij}$ between $X_{m}\left(i\right)$ and $X_{m}\left(j\right)$ through a fuzzy function:(58)$$D_{m,ij}=\mu \left(d_{m,ij},n,r\right)$$
where the fuzzy function $\mu (d_{m,ij},n,r)$ is the exponential function
(59)$$\mu \left(d_{m,ij},n,r\right)=exp\left(-\frac{d_{ij}^{n}}{r}\right).$$

For each vector $X_{m}\left(i\right)$, averaging all the similarity degree of its neighboring vectors $X_{m}\left(j\right)$, we get:(60)$$B_{m}\left(n,r,i\right)=\frac{1}{N-m}\sum_{j=1,j\neq i}^{N-m}D_{m,ij}.$$

Determine the function $B_{m}\left(n,r\right)$ as:(61)$$B_{m}\left(n,r\right)=\frac{1}{N-m}\sum_{i=1}^{N-m}B_{m}\left(n,r,i\right).$$

Similarly, form the vector sequence $\left\{X_{m+1}\left(i\right)\right\}$ and get the function $A_{m}\left(n,r\right)$:(62)$$A_{m}\left(n,r,i\right)=\frac{1}{N-m}\sum_{j=1,j\neq i}^{N-m}D_{m+1,ij}$$
(63)$$A_{m}{\left(n,r\right)}^{=}\frac{1}{N-m}\sum_{i=1}^{N-m}A_{m}\left(n,r,i\right).$$

Finally, we can define the parameter FuzzyEn(m,n,r) of the sequence as the negative natural logarithm of the deviation of $B_{m}\left(n,r\right)$ from $A_{m}\left(n,r\right)$:(64)$$FuzzyEn\left(m,n,r\right)=\lim_{N\to \infty }\left(lnB_{m}\left(n,r\right)-lnA_{m}\left(n,r\right)\right)$$
which, for finite databases, can be estimated from the statistic:(65)$$FuzzyEn\left(m,n,r,N\right)=lnB_{m}\left(n,r\right)-lnA_{m}\left(n,r\right)=-ln\left(\frac{A_{m}\left(n,r\right)}{B_{m}\left(n,r\right)}\right).$$

There are three parameters that must be fixed for each calculation of FuzzyEn. The first parameter *m*, as in ApEn and SampEn, is the length of sequences to be compared. The other two parameters, *r* and *n*, determine the width and the gradient of the boundary of the exponential function respectively.

#### 2.9.4. Modified Sample Entropy

The measure SampEn may have some problems in the validity and accuracy because the similarity definition of vectors is based on the Heaviside function, of which the boundary is discontinuous and hard. The sigmoid function is a smoothed and continuous version of the Heaviside function.

In 2008, a modified sample entropy (mSampEn), based on the nonlinear Sigmoid function, was proposed to overcome the limitations of SampEn [17]. The mSampEn is similar to FuzzyEn, the only differences is that instead of Equation (Equation 59), mSampEn uses the fuzzy membership function:(66)$$D_{m,ij}=\mu \left(d_{m,ij},r\right)=\frac{1}{1+exp[\left(d_{m,ij}-0.5\right)/r]}.$$

#### 2.9.5. Fuzzy Measure Entropy

In 2011, based on FuzzyEn definition, the fuzzy measure entropy (FuzzyMEn) was proposed by Liu and Zhao [122]. FuzzyMEn combines local and global similarity in a time-series and allows a has discrimination for time-series with different inherent complexities. It is defined as:(67)$$FuzzyMEn\left(m,r_{L},r_{G},n_{L},n_{G},N\right)=FuzzyEn\left(m,n_{L},r_{L},N\right)+FuzzyEn\left(m,n_{G},r_{G},N\right)$$
where $FuzzyEn(m,n_{L},r_{L},N)$ and $FuzzyEn(m,n_{G},r_{G},N)$ are obtained by Equation (Equation 65) considering in the Equation (Equation 55) the local vector sequence $XL_{m}\left(i\right)$ and global vector sequence $XG_{m}\left(i\right)$:(68)$$XL_{m}\left(i\right)=X_{m}\left(i\right)=\left\{x_{i},x_{i+1},x_{i+2},\ldots ,x_{i+m-1}\right\}-x_{0}\left(i\right)$$
(69)$$XG_{m}\left(i\right)=\left\{x_{i},x_{i+1},x_{i+2},\ldots ,x_{i+m-1}\right\}-\bar{x}.$$

The vector $XL_{m}\left(i\right)$ represents *m* consecutive x(i) values but removing the local baseline $x_{0}\left(i\right)$, which is defined in Equation (Equation 56). The vector $XG_{m}\left(i\right)$ also represents *m* consecutive x(i) values but removing the global mean value $\bar{x}$ which is defined as:(70)$$\bar{x}=\frac{1}{N}\sum_{i=1}^{N}x_{i}.$$

#### 2.9.6. Kernel Entropies

In 2005, Xu et al. proposed another modification of ApEn, the approximate entropy with Gaussian kernel [19]. It exploits the fact that the Gaussian kernel function can be used to give greater weight to nearby points by replacing the Heaviside function, in Equation (Equation 48), by k(i,j,r). The first kernel proposed k(i,j,r) is defined as:(71)$$k\left(i,j,r\right)=exp\left(-\frac{{∥x(i)-x(j)∥}^{2}}{10r^{2}}\right)$$
and the kernel-based entropy (KbEn) [123] is given as:(72)$$KbEn\left(m,r\right)=\Phi_{m}^{r}-\Phi_{m+1}^{r}$$
where $\Phi_{m}^{r}$ was defined in Equation (Equation 49).

Therefore, if $k\left(i,j,r\right)=\theta \left(max_{k=1,\ldots ,m}\left|x\left(i+k-1\right)-x\left(j+k-1\right)\right|-r\right)$ and θ is the Heaviside function then the resulting entropy value is the ApEn. The same procedure of changing the distance measure can be applied to define the sample entropy with Gaussian kernel [61].

In 2015, Mekyska et al. [124] proposed 6 other kernels based in approximate and sample entropies: exponential kernel entropy (EKE):(73)$$k\left(i,j,r\right)=exp\left(-\frac{∥x(i)-x(j)∥}{2r^{2}}\right);$$

Laplacian kernel entropy (LKE):(74)$$k\left(i,j,r\right)=exp\left(-\frac{∥x(i)-x(j)∥}{r}\right);$$
circular kernel entropy (CKE):(75)$$k\left(i,j,r\right)=\frac{2}{\pi }\arccos\left(-\frac{∥x(i)-x(j)∥}{r}\right)-\frac{2}{\pi }\frac{∥x(i)-x(j)∥}{r}\sqrt{1-{\left(\frac{∥x(i)-x(j)∥}{2r^{2}}\right)}^{2}}$$
for $∥x(i)-x(j)∥<r$, and zero otherwise; spherical kernel entropy (SKE):(76)$$k\left(i,j,r\right)=1-\frac{3}{2}\frac{∥x(i)-x(j)∥}{r}+\frac{1}{2}{\left(\frac{∥x(i)-x(j)∥}{2r^{2}}\right)}^{3}$$
for $∥x(i)-x(j)∥<r$, and zero otherwise; Cauchy kernel entropy (CauchyKE):(77)$$k\left(i,j,r\right)=\frac{1}{1+\frac{{∥x(i)-x(j)∥}^{2}}{r}}$$
for $∥x(i)-x(j)∥<r$, and zero otherwise; triangular kernel entropy (TKE):(78)$$k\left(i,j,r\right)=1-\frac{\left|x(i)-x(j)\right|}{r}$$
for $\left|x(i)-x(j)\right|<r$, zero otherwise.

Zaylaa et al. [123], called Gaussian entropy, exponential entropy, circular entropy, spherical entropy, Cauchy entropy and triangular entropy to kernels entropies based in ApEn.

### 2.10. Multiscale Entropy

The multiscale entropy approach (MSE) [60,125] was inspired by Zhang’s proposal [126], and considers the information of a system’s dynamics on different time scales. Multiscale entropy algorithms are composed of two main steps. The first one is the construction of the time-series scales: using the original signal, a scale, *s*, is created from the original time-series, through a coarse-graining procedure, i.e., replacing *s* non-overlapping points by their average. In the second step, the entropy (sample entropy, permutation entropy, fuzzy entropy, dispersion entropy, and others) is computed for the original signal and for the coarse-grained time-series to evaluate the irregularity for each scale. For this reason, methods of multiple scale entropy (such as entropy of entropy [127], composite multiscale entropy [128], refined multiscale entropy [129], modified multiscale entropy [130], generalized multiscale entropy [131], multivariate multiScale entropy [132], and others) were not explored in this paper.

## 3. The Entropy Universe Discussion

*The Entropy Universe* is presented in Figure 10.

Entropy is the uncertainty of a single random variable. We can define conditional entropy H(X|Y), which is the entropy of a random variable conditional on another random variable’s knowledge. The reduction in uncertainty due to the knowledge of another random variable is called mutual information.

The notion of Kolmogorov entropy as a particular case of conditional entropy includes various entropy measures, and estimates proposed to quantify the complexity of a time-series aimed at the degree of predictability of the underlying process. Measures such as approximate entropy, sample entropy, fuzzy entropy, and permutation entropy are prevalent for estimating Kolmogorov entropy in several fields [65].

Other measures of complexity are also related to entropy measures such as the Kolmogorov complexity. The original ideas of Kolmogorov complexity were put forth independently and almost simultaneously by Kolmogorov [133], Solomonoff [134], and Chaitin [135]. Teixeira et al. [136] studied relationships between Kolmogorov complexity and entropy measures. Consequently, in Figure 10 a constellation stands out that relates these entropies.

The concept of entropy has a complicated history. It has been the subject of diverse reconstructions and interpretations, making it difficult to understand, implement, and interpret. The concept of entropy can be extended to continuous distributions. However, as we mentioned, this extension can be very problematic.

Another problem we encountered was the use of the same name for different measures of entropy. The quadratic entropy was defined by Lake [87] and by Rao [89], but despite having the same name, the entropies are very different. The same is valid with graph entropy, which was defined differently by Mowshowitz [102] and by János Körne [106]. Different fuzzy entropies have been proposed over time, as mentioned in Section 2.9.3, but always using the same name. On the other hand, we find different names for the same entropy. It is common in the literature to find thermodynamic entropy, differential entropy, metric entropy, Rényi entropy, geometric entropy, and quantum entropy with different names (see Figure 10). Moreover, the same name is used for a family of entropies and a particular case. For example, the term Rényi entropy is used for all Rényi entropy families and to the particular case q=2.

It is also common to have slightly different definitions of some entropies. The Tsallis entropy formula in some papers appears with *k*, but in other papers, the authors consider k=1, which is not mentioned in the text.

Depending on the application area, different log bases are used for the same entropy. In this paper, in the Shannon entropy formula, we use the natural logarithm, and in the Rényi entropy formula, we use base 2 logarithm. However, it is common to find these entropies defined with other logarithmic bases in the scientific community, which needs to be considered when relating entropies.

In this paper, we study only single variables, but many other entropies have not been addressed. Nonetheless, some entropies were created to measure the information between two or more time-series. The relative entropy (also called Kullback–Leibler divergence) is a measure of how one probability distribution is different from a second, reference probability distribution. The cross-entropy is an index for determining the divergence between two sets or distributions. Many other like, conditional entropy, mutual information, information storage [65], relative entropy, will not be covered in this paper. While belonging to the same universe, they should be considered in a different galaxy.

Furthermore, in the literature, we find connections between the various entropies and other dispersion and uncertainty measures. Ronald Fisher defined a quantity called Fisher information as a strict limit on the ability to estimate the parameters that define a system [137,138]. Several researchers have shown that there is a connection between Fisher’s information and Shannon’s differential entropy [46,139,140,141]. The relationship between the variance and entropies was also explored in many papers such as [141,142,143]. Shannon defined what he called the derived quantity of entropy power (also called entropy rate power) to be the power in a Gaussian white noise limited to the same band as the original ensemble and having the same entropy. The entropy power defined by Shannon is the minimum variance that can be associated with the Gaussian differential entropy [144]. However, the entropy power can be calculated associated with any given entropy. The in-depth study of the links between *The Entropy Universe* and other measures of dispersion and uncertainty remains as future work. As well as the exploration of the possible log-concavity of these measures along with the heat flow, for example, as recently established for Fisher information [145].

Like our universe, the universe of entropies can be considered infinite because it is continuously expanding.

## 4. Entropy Impact in the Scientific Community

The discussion and the reflection on the definitions of entropy and how entropies are related are fundamental. However, it is also essential to understand each entropy’s impact in the scientific community and in which areas each entropy applies. The main goals of this section are two-fold. The first is to analyze the number of citations for each entropy in recent years. The second is to understand the application areas of each entropy discussed in the scientific community.

For the impact analysis, we have used the Web of Science (WoS) and Scopus databases. Those databases provide access to the analysis of citations of scientific works from all scientific branches and have become an important tool in bibliometric analysis[146]. On both websites, researchers and the scientific community can access databases, analysis, and insights. Several researchers have studied the advantages and limitations of using the WoS or Scopus databases [147,148,149,150,151].

The methodology used was the following in both databases: we searched for the paper’s title in which entropy was proposed and collected the year publication, the number of citations total and by year, for the last 16 years. In WoS, we selected in *Research Areas* the ten areas with a higher record count, and in Scopus, we selected the ten *Documents by subject area* with a larger record count for each entropy.

### 4.1. Number of Citations

In Table 1, we list the paper in which each entropy was proposed. We also present the number of citations of the paper in Scopus and WoS since the paper’s publication.

In general, as expected, the paper that firstly defined each entropy of the universe has more citations in Scopus than in WoS. Based on the number of citations in the two databases, the five most popular entropies in the scientific community are the Shannon/differential, maximum, Tsallis, SampEn, and ApEn entropies. Considering the papers published until 2000, only the papers that proposed the Boltzmann, the Boltzmann-Gibbs-Shannon, the minimum, the geometric, and the tone-entropy entropies have less than one hundred citations in each database. The papers that proposed the empirical mode decomposition energy entropy and the coefficient of sample entropy are the most cited ones among the entropies introduced in the last sixteen years (see Table 1). Of the entropies proposed in the last five years, the dispersion entropy paper was the most cited one, followed by the one introducing the kernels entropies.

Currently, the WoS platform only covers the registration and analysis of papers after 1900. Therefore, we have not found all the documents needed for the analysis to be developed in this section. Ten papers were not found in the WoS, as shown in Table 1. In particular, entropies’ papers widely cited in the scientific community were not found, such as Gibbs, Quantum, Rényi, and Fuzzy entropies. Currently, Scopus has more than 76.8 million main records: 51.3 million records after 1995 with references and 25.3 million records before 1996, the oldest from 1788 [152]. We found the relevant information regarding all papers that proposed each entropy in Scopus.

Next, we analyzed the impact of each entropy on the scientific community in recent years. Figure 11 and Figure 12 show the number of citations of the paper proposing each entropy in the last 16 years (from 2004 to 2019), respectively, in the Scopus and WoS databases. Note that the colors used in the Figure 11 and Figure 12 are in accordance with the colors in Figure 1 and we did not use the 2020 information once we considered that those values were incomplete at the research time. The range of citations of the papers that introduced each entropy is extensive; therefore, we chose to use a logarithmic scale in Figure 11 and Figure 12. For entropies whose paper that introduced it has been cited more than three times in the past 16 years, we have drawn the linear regression of the number of citations as a function of the year. In the legends of Figure 11 and Figure 12, we present the slope of the regression line, b, and the respective *p*-value. For p<0.05 the slope was considered significant. Shannon/Differential paper is the most cited in the last sixteen years, as shown in Figures and Table 1.

In recent years, Shannon/differential, sample, maximum, Tsallis, permutation/sorting, and approximate were the entropies most cited, as we can see in both Figures. However, Rényi entropy joins that group in Figure 12. Fluctuation-based dispersion entropy was proposed in 2018, so we only have citations in 2019 that correspond to the respective values in Table 1.

In the last sixteen years, there were several years in which Boltzmann-Gibbs-Shannon, graph, minimum, spectral, kernel-based, Tsallis permutation entropy, Δ−entropy, and fuzzy measure entropies had not been cited, according to WoS information (Figure 11). While, Boltzmann, Boltzmann-Gibbs-Shannon, minimum, geometric, Δ−entropy, kernel-based, Tsallis permutation, and fuzzy measure entropies papers have not been cited in all years as reported in Scopus (Figure 12). In particular, the paper that introduced Boltzmann-Gibbs-Shannon entropy, with the least number of citations in the last sixteen years, had only one quote in 2012 on the WoS platform, and in Scopus, it had three citations (2005, 2010, and 2012).

Figure 11 shows that the number of citations of the paper proposing sample entropy has increased significantly and is the fastest: the number of citations per year went from 13 in 2004 to 409 in 2019. The permutation/sorting paper had an increasing number of citations as well; the number of citations per year increased significantly from 3 in 2004 to 300 in 2019. From this information, we infer that these entropies have increased their importance in the scientific community. Based on the WoS database, the two entropies proposed in the last eight years, whose paper has strictly increased the number of citations, are dispersion and bubble entropies, respectively, from 2 in 2016 to 32 in 2019 and 1 in 2017 to 11 in 2019. However, from the ones proposed in the last eight years, the two most cited are dispersion and kernels entropies’ papers. Regarding the paper that introduced the wavelet entropy, and according to Figure 11, the year with the most citations was 2015. Since then, the number of citations per year has been decreasing.

The results from Scopus, displayed in Figure 12, show that among the entropy articles where the number of citations increased significantly are the Shannon/differential (from 754 in 2004 to 2875 in 2019) and the sample (from 16 in 2004 to 525 in 2019). More recently, the number of citations of the papers that introduced dispersion and bubble entropies continuously increased from 1 in 2016 to 42 in 2019 and from 3 in 2017 to 13 in 2019, respectively. In 2014, there were 12 entropy papers more cited than the sample entropy paper, but in 2019 the sample entropy paper was the second most cited. This progress implies that the impact of SampEn on the scientific community has increased significantly. On the other hand, the spectral entropy paper lost positions in 2004, it was the 6th most cited, and moved to position 16 in 2019 (see Figure 12). The results obtained on the WoS and Scopus platform on the impact of entropies on scientific communities complement and reinforce each other.

### 4.2. Areas of Application

To understand each entropy’s areas of application in the scientific community, we consider the ten areas that most cited the paper that introduced the entropy, according to the largest number of records obtained in Web of Science and the Scopus databases. The results are displayed in Figure 13 and Figure 14, respectively. The entropies are ordered in the chronological order of the work that proposed them, meaning the first line corresponds to the oldest entropy paper. There are ten color tones in the figures representing the range from 0 (blue) to 10 (red), where 0 is assigned to the research area that least mentioned the paper that introduced the entropy. In contrast, 10 is assigned to the research area that most cited the paper.

Note that there are papers in which the entropies were proposed that do not have ten application areas in the databases used. For example, the Rényi permutation entropy has only nine areas in the WoS, and the ranked-based entropy has only six areas of application in Scopus.

Figure 13 and Figure 14 show that, over the years, the set of areas that apply entropies is increasingly numerous. However, we did not obtain the same results for the two figures.

Based on the ten *Research Areas* with the largest number of records obtained on the WoS, we obtained 45 *Research Areas* (Figure 13).

Initially, the research areas that most cited the entropy papers were Physics, Mathematics, and Computer Science. However, according to data from the WoS, there is a time interval between 1991 (approximate entropy) and 2007 (intrinsic model entropy) in which the emerging entropies have more citations in the area of Engineering. Of the thirty works covered by the WoS database, sixteen were most cited in the Engineering area, nine in the Physics area, five in the Computer Science area, and two in the Mathematics area. The second area of research in which the paper that introduced tone-entropy was most cited is Psychology. The second area in which the paper that proposes Tsallis permutation entropy was most cited is Geology. According to the WoS database, the papers that introduced entropies were little cited in medical journals.

Figure 14, based on the top 10 *Documents by subject area* of Scopus, covers all papers that proposed each entropy described in this paper. However, this top 10 contains about half of the *Research Areas* provided on the WoS (25 *Documents by area subject*). Also, we observe that the set of research areas that most cite the works is broader. Of the forty works covered by the Scopus database, the twelve most cited papers are in the area of Computer Science, eight in the area of Physics and Astronomy, seven in the area of Engineering, seven in the area of Mathematics, three in the area of Medicine, one in the area of Chemistry and two papers with the same number of citations in different areas. Over the years, the papers that introduced entropies have been most cited in medical journals by the Scopus database.

According to the Scopus platform, areas such as Business, Management and Accounting, and Agricultural and Biological Sciences are part of the ten main areas that cite the papers that proposed the universe’s entropies. When we did the data collection on the WoS, the paper that introduced minimum entropy was mentioned by papers from only four areas, but it is complete in the top 10 of Scopus. On the WoS, the papers that proposed approximate entropy, tone-entropy, and coefficient of sample entropy are the most cited in the Engineering area. In contrast, in the Scopus database, they are most cited in the medical field.

There are several differences between the results obtained in the application areas of the two platforms. However, we also find similarities. Shannon/differential, maximum, topological, graph, minimum, tone-entropy, wavelet, EMDEnergyEn, intrinsic mode, Rényi permutation, kernels, dispersion and fluctuation-based dispersion entropies’ introductory papers show the same application areas with more citations in Scopus and WoS database.

We believe that the differences found may be due to the number of citations in the papers and the list of areas for each platform being different. Therefore, it is important to show the results obtained based on the WoS and Scopus since, when the results are the same, it strengthens, while when the results are different, the two analyzes complement each other.

## 5. Conclusions

In this paper, we introduce *The Entropy Universe*. We built this universe by describing in-depth the relationship between the most applied entropies in time-series for different scientific fields, establishing bases for researchers to properly choose the variant of entropy most suitable for their data and intended objectives. We have described the obstacles surrounding the concept of entropy in scientific research. We aim that this work help researchers choosing the best entropy for their time-series analysis. Among the problems, we discussed and reflected on the extension of the concept of entropy from a discrete variable to a continuous variable. Also, we point out that different entropies have the same name and the same entropies have different names.

The papers that proposed entropies have been increasing in the number of citations, and the Shannon/differential, Tsallis, sample, permutation, and approximate had been the most cited entropies. Permutation/sorting entropy were the ones that most increased the impact on scientific works, in the last sixteen years. Of the entropies proposed in the past five years, kernels and dispersion entropies are the ones that have the greatest impact on scientific research. Based on the ten areas, with the largest number of records of the paper that introduced each new entropy, obtained from WoS and Scopus, the areas that most applied the entropies are Computer Science, Physics, Mathematics, and Engineering. However, there are differences between the results achieved by the two databases. According to the WoS database, the papers that introduced the entropies were rarely cited in medical journals. However, we did not obtain the same result from the Scopus database.

*The Entropy Universe* is an ongoing work since the number of entropies is continually expanding.

## Figures and Tables

**Figure 1 entropy-23-00222-f001:**
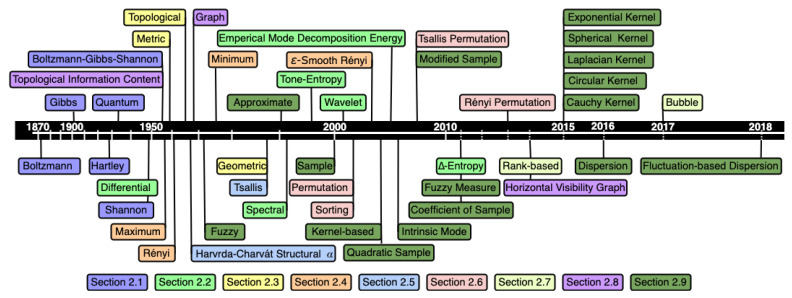
Timeline of the universe of entropies discussed in this paper. Timeline in logarithmic scale and colors refer to the section in which each entropy is defined.

**Figure 2 entropy-23-00222-f002:**
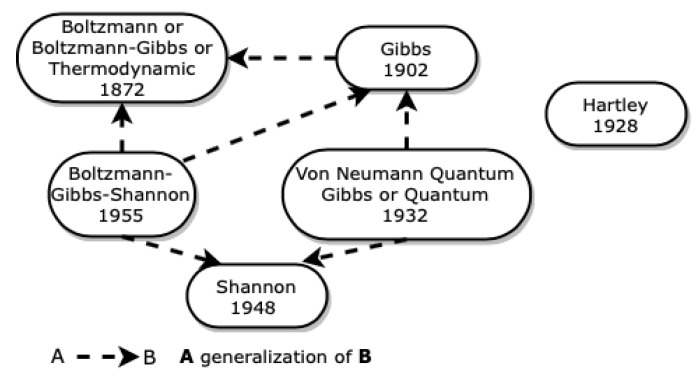
Origin of The Entropy Universe.

**Figure 3 entropy-23-00222-f003:**
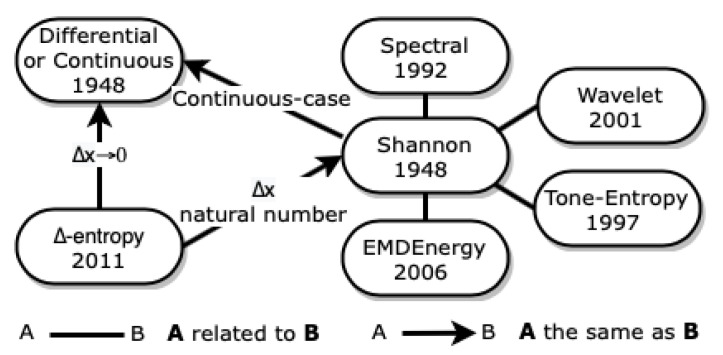
Entropies related to Shannon entropy.

**Figure 4 entropy-23-00222-f004:**
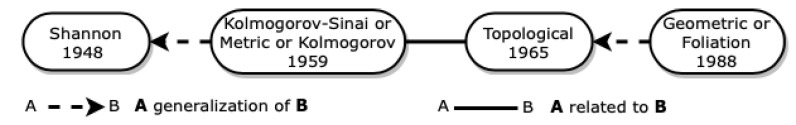
Relationship between Kolmogorov, topological and geometric entropies.

**Figure 5 entropy-23-00222-f005:**
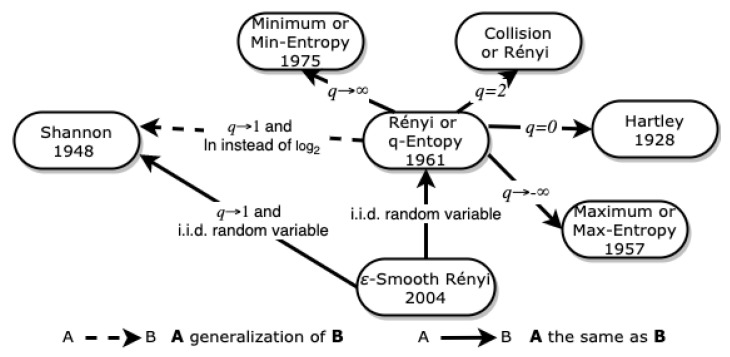
Relationship between Rényi entropy and its particular cases.

**Figure 6 entropy-23-00222-f006:**
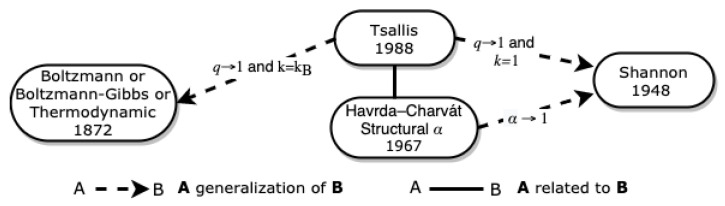
Relationship between Havrda–Charvát structural α-entropy Tsallis and others entropies.

**Figure 7 entropy-23-00222-f007:**
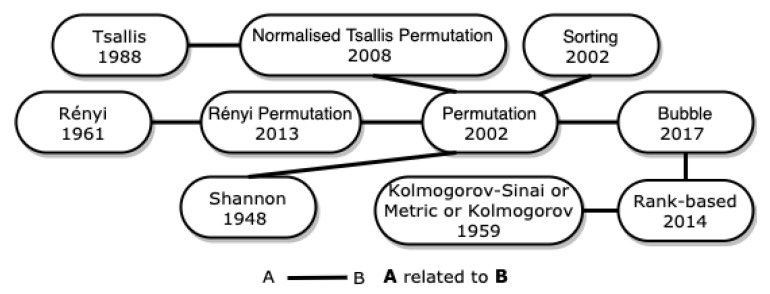
Entropies related to permutation entropy.

**Figure 8 entropy-23-00222-f008:**
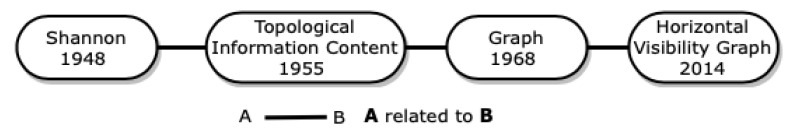
Relations between topological information content, graph entropy and horizontal visibility graph entropy.

**Figure 9 entropy-23-00222-f009:**
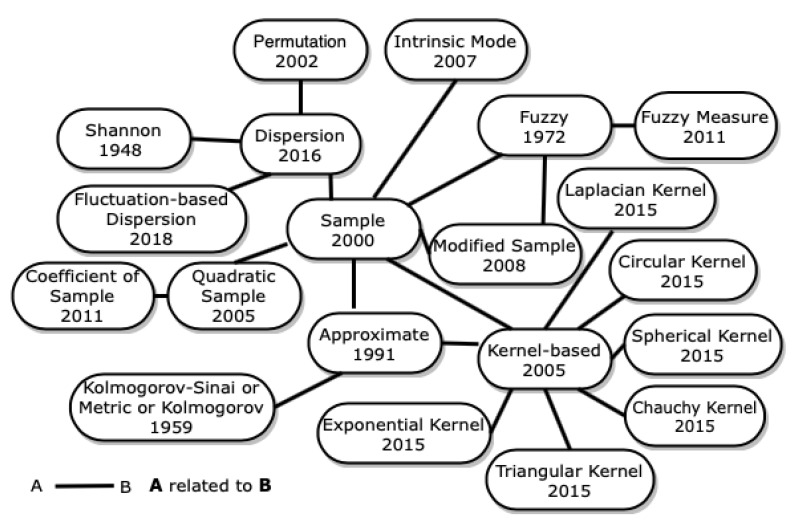
Entropies related to sample entropy.

**Figure 10 entropy-23-00222-f010:**
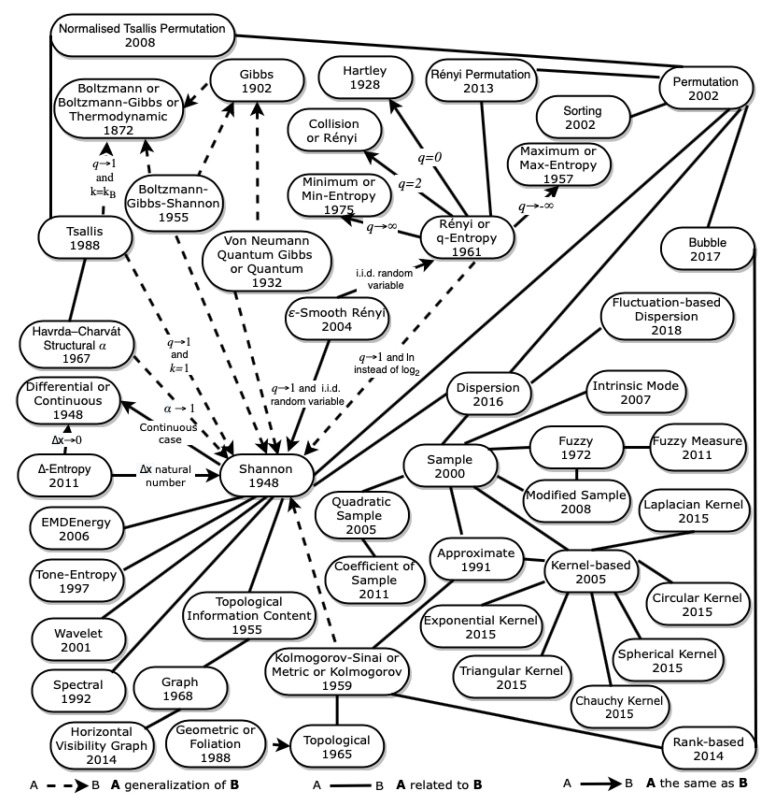
The Entropy Universe.

**Figure 11 entropy-23-00222-f011:**
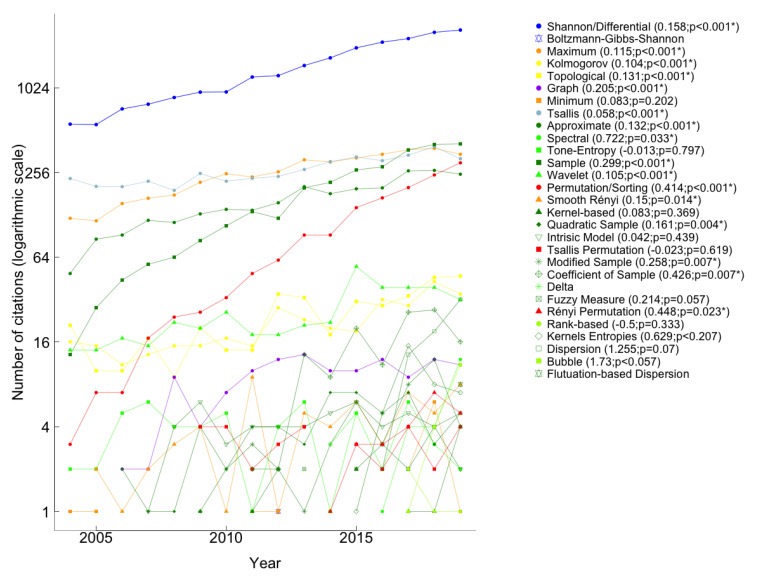
Number of citations by year in the WoS between 2004 and 2019 of the papers proposing each measure of entropy, in logarithmic scale ($log_{2}\left(\mathrm{Number}\;\mathrm{of}\;\mathrm{citations}\right)$). In the legend, the ordered pair (β, *p*-value), in papers cited in more than three years, corresponds to the slope of the regression line, β, and the respective *p*-value. Statistically significant slopes (p<0.05) are marked with *.

**Figure 12 entropy-23-00222-f012:**
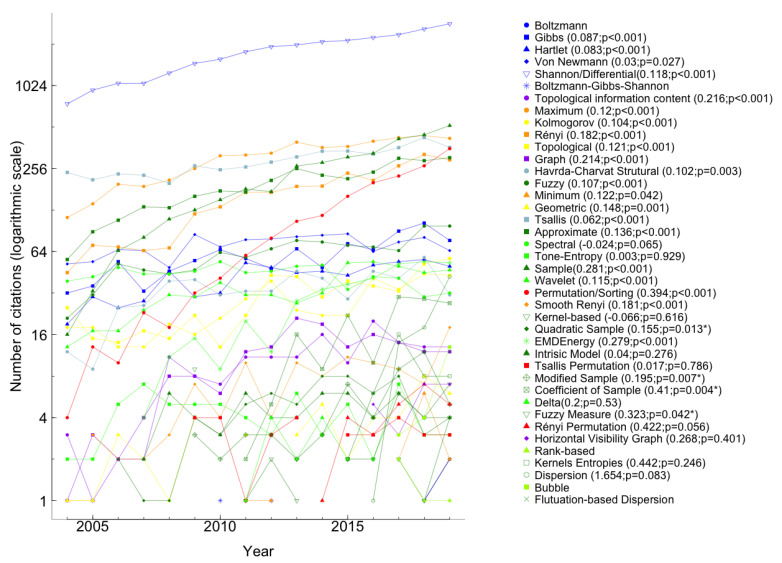
Number of citations by year in Scopus between 2004 and 2019 of the papers proposing each measure of entropy, in logarithmic scale ($log_{2}\left(\mathrm{Number}\;\mathrm{of}\;\mathrm{citations}\right)$). In the legend, the ordered pair (β, *p*-value), in papers cited in more than three years, corresponds to the slope of the regression line, β, and the respective *p*-value. Statistically significant slopes (p<0.05) are marked with *.

**Figure 13 entropy-23-00222-f013:**
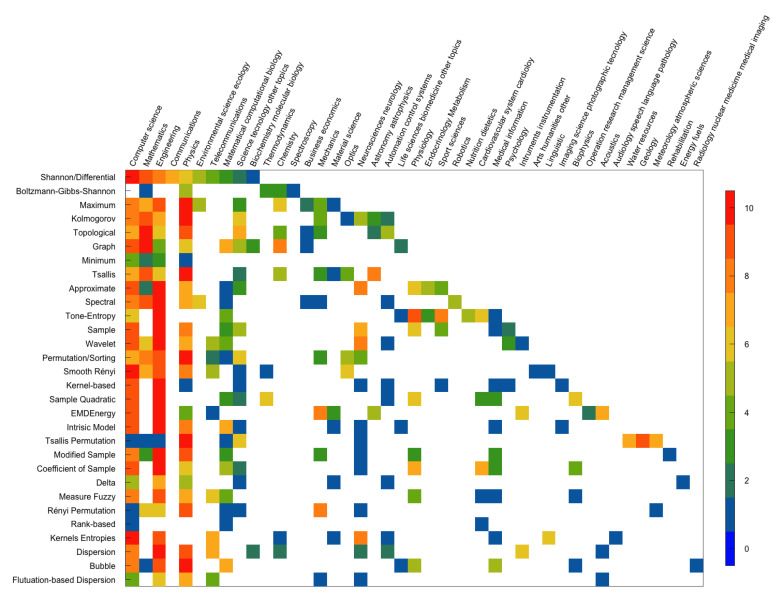
The ten areas that most cited each paper introducing entropies according to the *Research Areas* of the WoS. Legend: range 0 (research area least cited)-10 (research area most cited).

**Figure 14 entropy-23-00222-f014:**
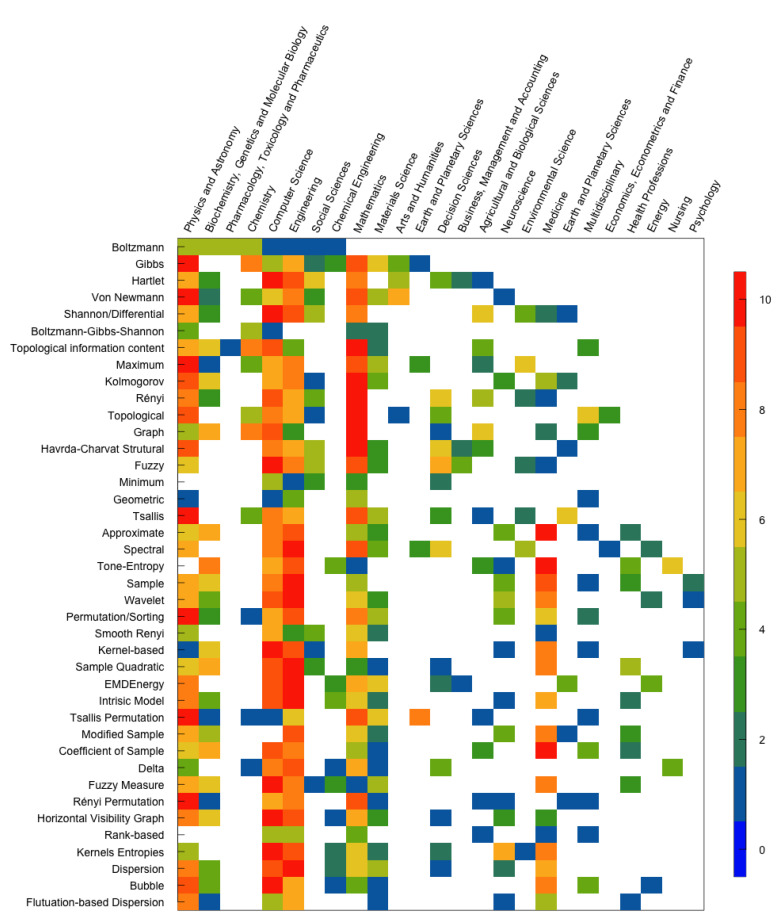
The ten areas of most cited papers that introduced entropies according to the *Documents by subject area* of the Scopus. Legend: range 0 (research area least cited)-10 (research area most cited).

**Table 1 entropy-23-00222-t001:** Reference and number of citations in Scopus and WoS of the paper that presented each entropy.

Name of Entropy	Reference	Year	Scopus	Web of Science
Boltzmann entropy	[25]	1900	5	-
Gibbs entropy	[26]	1902	1343	-
Hartley entropy	[29]	1928	902	-
Von Newmann entropy	[30]	1932	1887	-
Shannon/differential entropies	[33]	1948	34,751	32,040
Boltzmann-Gibbs-Shannon	[42]	1955	8	7
Topological information content	[100]	1955	204	-
Maximum entropy	[83]	1957	6661	6283
Kolmogorov entropy	[55]	1958	693	662
Rényi entropy	[79]	1961	3149	-
Topological entropy	[66]	1965	728	682
Havrda–Charvát structural α-entropy	[90]	1967	744	-
Graph entropy	[102]	1968	207	195
Fuzzy entropy	[118]	1972	1395	-
Minimum entropy	[84]	1975	22	17
Geometric entropy	[74]	1988	71	-
Tsallis entropy	[91]	1988	5745	5467
Approximate entropy	[112]	1991	3562	3323
Spectral entropy	[49]	1992	915	26
Tone-entropy	[51]	1997	85	76
Sample entropy	[15]	2000	3770	3172
Wavelet entropy	[52]	2001	582	465
Permutation/sorting entropies	[13]	2002	1900	1708
Smooth Rényi entropy	[86]	2004	112	67
Kernel-based entropy	[19]	2005	15	13
Quadratic sample entropy	[87]	2005	65	68
Empirical mode decomposition energy entropy	[53]	2006	391	359
Intrinsic mode dispersion entropy	[115]	2007	59	55
Tsallis permutation entropy	[92]	2008	35	37
Modified sample entropy	[17]	2008	58	51
Coefficient of sample entropy	[16]	2011	159	136
Δ−entropy	[54]	2011	13	10
Fuzzy entropy	[122]	2011	23	18
Rényi permutation entropy	[14]	2013	28	26
Horizontal visibility graph entropy	[109]	2014	22	-
Rank-based entropy	[94]	2014	6	6
Kernels entropies	[124]	2015	46	39
Dispersion entropy	[18]	2016	98	84
Buble entropy	[95]	2017	25	21
Fluctuation-based dispersion entropy	[116]	2018	16	10
Legend: -paper not found in database.				

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
