# Peer review of "The Entropy Universe"

_entropy, 2021, doi:10.3390/e23020222_

Round 1

Reviewer 1 Report

The paper reviews the properties and use of a plethora of different entropy formulations. It is a well written paper and provides a very comprehensive and useful tool for researchers in applied mathematics, physics, and engineering areas, among others.

I believe the draft could also highlight the connection with other measures of dispersion and uncertainty, even if not formally associated with an entropy classification. I am referring to simple measures as the variance, the entropy power, the Fisher information, and divergence measures such as the mutual information, (and the controversial lautum information), the Kullbach-Leibler divergence, and some adaptive, flexible and multiscale entropy formulations (e.g., Costa, M. et al., Multiscale Entropy Analysis of Complex Physiologic Time Series, Physical Review Letters, 89, 6, Aug. 5, 2002).

Also of interest is the exploration of the possible log-concavity of these measures along the heat flow, e.g., as recently established for Fisher information (cf, Ledoux, M. et al., Log-convexity of Fisher information along heat flow, arxiv draft). These properties entail the specific entropy formulations with very useful characteristics, such as isoperimetric inequalities and other key attributes that can be useful for establishing bounds and phase transitions. Shannon's entropy power inequality is a fundamental example of these kinds of properties.

I do not propose a significant reorganization of the paper but I believe these remarks may be included as a few paragraphs of discussion and may add to the usefulness of the manuscript.

Reviewer 2 Report

Authors should develop statistical models to interpret the data presented in Figure 11 and 12. They need to establish that the growth they are claiming in different forms of entropy are statistically significant. A simple model of citations as a function of trend line may suffice to understand the trend and the growth in different variations of entropy. 

Reviewer 3 Report

Dear author,

The review work that I have read and verified is completely original, the work is well justified and I consider it to be of great interest to ENTROPY readers. The methodology used is correct and pertinent in revision-type work. I have not identified errors or omissions, I am sure, in my professional opinion, that the work is ready to be published.

Best regards,

Professor Abdiel Ramírez-Reyes
